# Predicting Restroom Dirtiness Based on Water Droplet Volume Using the LightGBM Algorithm

**DOI:** 10.3390/s25072186

**Published:** 2025-03-30

**Authors:** Sumio Kurose, Hironori Moriwaki, Tadao Matsunaga, Sang-Seok Lee

**Affiliations:** 1School of Engineering, Tottori University, 4-101 Koyama-Minami, Tottori 680-8552, Japan; sumiokurose@yahoo.co.jp (S.K.);; 2Asuka Bisou Co., Ltd., 296-1 Daigo, Kashihara 634-0072, Japan; hironori.moriwaki@asukabiso.co.jp

**Keywords:** restroom dirtiness, LightGBM, data augmentation techniques, water droplet volume prediction, cleaning schedule prediction

## Abstract

This study examines restroom cleanliness in public facilities, department stores, supermarkets, and schools by using water droplet volumes around washbowls as an indicator of usage. Rising cleaning costs due to labour shortages necessitate more efficient restroom maintenance. Quantifying water droplet accumulation and predicting cleaning schedules can help optimise cleaning frequency. To achieve this, water droplet volumes were measured at specific time intervals, with significant variations indicating increased restroom usage and potential dirt buildup. For real-world assessment, acrylic plates were placed on both sides of washbowls in public restrooms. These plates were collected every hour over five days and analysed using near-infrared photography to track changes in water droplet areas. The collected data informed the development of a prediction system based on the decision tree method, implemented via the LightGBM framework. This paper presents the developed prediction system, which utilises in situ water droplet volume measurements, and evaluates its accuracy in forecasting restroom cleaning needs.

## 1. Introduction

### 1.1. Current Issues Faced by Cleaning Companies

Restrooms in public and commercial facilities, such as department stores, supermarkets, and schools, are typically maintained by cleaning companies. However, cleaning is often not considered a value-adding service, leading to cost compromises. Recently, rising labour costs have further increased cleaning expenses, prompting companies to focus on efficiency by optimizing cleaning frequency and duration.

Typically, commercial and public restrooms are cleaned before opening to the public, followed by periodic inspections. During these inspections, consumables such as toilet paper and hand soap are restocked, and any visible dirt or stains are addressed to maintain cleanliness [1,2,3,4,5]. This additional effort ensures that restrooms remain hygienic and well maintained.

### 1.2. Solution for Restroom Cleaning Issues

Historically, public restroom cleanliness has received minimal attention. However, restrooms in restaurants, department stores, and supermarkets are now more closely monitored, as their cleanliness directly impacts the facility’s image. Recognizing this, businesses have increasingly prioritised restroom maintenance.

Figure 1 and Figure 2 present survey responses from 2000 participants regarding restroom cleanliness in restaurants. Figure 1 illustrates responses to the question ‘How do you feel about clean restaurant restrooms?’. Nearly 80% of respondents stated that clean restrooms improve a restaurant’s image. Figure 2 shows responses to the question ‘How do you feel about dirty restaurant restrooms?’. Here, 77% of respondents felt that unclean restrooms negatively impact a restaurant’s image, indicating a lack of consideration for customers. These findings suggest that restroom cleanliness significantly influences public perception, not just in restaurants but also in department stores and supermarkets.

Figure 3 presents survey responses to the question ‘How comfortable do you feel using public restrooms?’. Participants, aged 20 to 60, included 80 respondents per sex per age group (totalling 800 respondents). Multiple answers were allowed [7].

Current cleaning routines are insufficient to ensure restroom cleanliness at all times. Setting an appropriate cleaning frequency and timing is essential [8,9]. Furthermore, if cleaning duration can be accurately predicted, restrooms can remain clean while minimizing cleaning costs.

As previously mentioned, assessing restroom dirtiness enables the determination of an optimal cleaning frequency and duration, allowing for a more flexible and efficient cleaning strategy. To achieve this, this study develops a prediction system and establishes a methodology for effective restroom maintenance.

## 2. Dirtiness Quantification Method

### 2.1. Definition of Dirtiness

Restrooms contain various sources of dirt, such as water droplets, hairs, and paper waste. While all types of dirt should be removed during cleaning, quantifying these different types of dirt presents challenges. Specifically, paper waste, dirt around urinals, and hairs in washbowls are difficult to measure precisely. To simplify the process, we defined the presence of water droplets around washbowls as an indicator of overall dirtiness.

Figure 4 displays images of water droplets placed using a digital pipette. The relationship between the number of pixels representing the water droplets and the corresponding volume was determined before the experiments, as shown in Figure 5. This relationship is approximated as linear.

For the in situ experiments, water droplets were collected on acrylic plates installed beside the washbowls. These droplets were then photographed, and the pixel-based images were converted to corresponding volumes using the method outlined in Section 2.2.

### 2.2. Overview of Method

The experiment followed a series of steps to determine the volume of water droplets around the washbowl. Water droplets were photographed using a near-infrared camera, as shown in Figure 6, where the droplets appear as black areas.

However, the image also contains extraneous black areas outside the acrylic plate, caused by background differences where no water droplets exist. To ensure accurate pixel counting, image-editing software (Photos, version 2024.11040.1002.0) was used to remove these irrelevant black areas. The cleaned-up bitmap image is shown in Figure 7. The number of pixels in the cleaned image (Figure 7) was then counted using the open-source software ImageJ (version 1.53). Finally, the pixel count was converted into the volume of water droplets using the established relationship shown in Figure 5.

This method enables the quantification of restroom dirtiness based on the volume of water droplets around the washbowl. While it does not provide a perfectly accurate estimate due to factors such as overlapping droplets, the method offers a reliable means to assess water droplet amounts. Additionally, it allows for the establishment of a cleanliness threshold, which is crucial for determining optimal cleaning timing.

## 3. In Situ Experiment and Results

### 3.1. Preparation of Water Droplet Images

To capture time-varying images of water droplets, experiments were conducted in the restrooms of a language school with approximately 70 students. The number of students present on any given day varied, and most students were from Asian countries studying Japanese for college entrance exams or Japanese qualification exams.

To collect water droplets around the washbowl, acrylic plates were placed on either side of the washbowl and replaced every hour for analysis. Figure 8 shows the actual setup, with transparent acrylic plates used in the experiment. The plates were colour-coded blue in the figure for better visibility.

Setting up a camera inside the restroom posed privacy concerns, and it was difficult to maintain the same camera position every hour. Moreover, distinguishing between male and female users was necessary. Consequently, the acrylic plates were collected every hour, and photographed in a separate room under consistent conditions, and the images were transferred to a computer connected to the camera. The camera lens was positioned 650 mm above the acrylic plates, as shown in Figure 9.

A near-infrared camera (NIRCam-640SN, Vision Sensing Co., Ltd., Osaka, Japan) was used for capturing the images, with specifications listed in Table 1. Since water absorbs at wavelengths of 1200, 1450, and 1940 nm in the near-infrared spectrum, a 1450 nm bandpass filter was attached to the camera to capture water droplets effectively. As a result, water droplets were depicted as black areas in the images.

Representative photographs taken during the in situ experiment are shown in Figure 10, Figure 11 and Figure 12. These images were captured at 10:30, 11:30, and 12:30 on the first day of the experiment at male restroom. The images on the left show the raw bitmap images, while the processed images on the right have had irrelevant black areas removed, as described in Section 2.2. Water droplet volumes were then estimated using the pre-determined relationship shown in Figure 5. These volume data were subsequently used as input for the prediction system.

### 3.2. Data Collection

Photographs of water droplets on the acrylic plates were captured at 1 h intervals, and the data were accumulated into ‘1 h before’ datasets. These images were collected over five days from both male and female restrooms. The data are presented in Figure 13 and Table 2. The variables lag1030 through lag1730 represent data collected at hourly intervals from 10:30 to 17:30. Additionally, the average temperature and humidity were recorded, as shown in Table 2.

The collected data on water droplet volumes exhibited significant variation, depending on individual usage, as shown in Figure 13. On some days, there were abrupt changes in droplet volumes in response to time variations. However, as the number of data points increased, this variation became less pronounced. Generally, the data indicated a tendency for the accumulated water droplet volume to increase with the number of restroom users. It is important to note that ventilation fans were continuously running in the restrooms, which led to the evaporation of water droplets over time. As a result, the observed water droplet volumes were slightly lower than the actual volumes, implying that the prediction may be somewhat underestimated.

### 3.3. Data Augmentation

As noted earlier, the data were collected for five days from both male and female restrooms but these were insufficient for training and prediction purposes. To overcome this limitation, a data augmentation method was applied, as long-term restroom experiments are constrained. Several data augmentation techniques have been reported in the literature, including the addition of random noise, data scaling, weighted dynamic time warping (DTW), dynamic time warping barycentric averaging (DBA) [11,12,13], moving block bootstrapping (MBB) [14,15,16,17], and the DoppelGANger framework based on generative adversarial networks (GAN) [18,19,20]. In anomaly detection, adding random noise is effective, and GANs can generate high-fidelity datasets. MBB generates pseudo-scattering through random sampling. However, our application required only a small portion of scattered datasets, and the datasets needed to be time-series. Therefore, we adopted the DBA method for our prediction model.

We primarily employed two methods for data augmentation:Averaging and scaling method: This approach involved successively taking averages of two nearest neighbouring data points, two second-nearest neighbouring data points, and so on. These averages were then multiplied by 0.9 and 1.1, and the process was repeated. This method generated three sets of augmented data by expanding the margins of the minimum and maximum values.Random number generation method: In this method, random numbers were generated between the maximum and minimum values for each dataset.

Based on the above, we primarily attempted the following two methods:

Details of the first method, involving averaging and scaling, are illustrated in Figure 14. This method resulted in a total of 60 datasets: five original datasets (data No. 1–5) and 55 augmented datasets (data No. 6–60). Each augmented dataset also included the average temperature and humidity for the day, obtained from a meteorological agency. In Figure 14, data No. 6 represents the average of data No. 1 and data No. 2, including temperature and humidity averages. Similarly, data No. 11 is the average of data No. 1 and data No. 3, and so on. By successively taking averages with two nearest neighbours, two second-nearest neighbours, and so on, we generated 20 augmented data points from the original five datasets.

Next, data No. 26 and data No. 27 were obtained by multiplying data No. 1 and data No. 2 by 1.1, respectively. Similarly, data No. 31 and data No. 33 were multiplied by 0.9. This step provided an additional 10 augmented datasets. The process was repeated by averaging the new data points, resulting in a total of 60 datasets. Figure 15 and Figure 16 show the actual water droplet data from the male restroom (Figure 13) and the augmented data from the male restroom, respectively.

The second method for data augmentation used random number generation. First, the maximum and minimum values from each hourly dataset were identified. The maximum value was multiplied by 1.1, and the minimum value by 0.9. Random numbers were then generated between these two values. This generated 55 random datasets, which, when combined with the five original datasets, resulted in a total of 60 datasets. To ensure consistency, the data were sorted in ascending order, as water droplet volumes could not decrease over time. Table 3 and Figure 17 display the augmented datasets generated by this method. Temperature and humidity were not incorporated, as the random number generation process is not time- or date-dependent.

Ultimately, three augmented datasets were prepared: (1) the averaging and scaling method, including temperature and humidity, (2) the averaging and scaling method without temperature and humidity, and (3) the random number generation method. These datasets were then used to predict water droplet volumes, and the accuracy of the prediction methods was evaluated.

### 3.4. Water Droplet Volume Prediction

We aimed to predict water droplet volumes at 17:30 based on data collected from 10:30 to 16:30. Specifically, our goal was to predict the water droplet volume for the hour following data collection. Although predicting the volume for 2 or 3 h intervals is more practical in real-world cleaning scenarios, this study focused on constructing and evaluating a model. Therefore, we initially chose to predict the volume for just 1 h after the data collection period.

For our predictions, we used the light gradient boosting machine (LightGBM) method [21,22,23,24,25,26], which operates within a decision tree framework. Specifically, we employed the DecisionTreeRegressor algorithm. Several other methods are available for time-series predictions, including recurrent neural networks (RNN) [27,28,29] and long short-term memory (LSTM) networks [30,31,32,33,34,35]. Both RNNs and LSTMs are based on neural networks: in RNNs, past time-series data are stored in hidden layers, whereas LSTMs store them in additional gates (forget, input, and output gates). These stored values are weighted and added to the input data.

We first evaluated the applicability and validity of the LightGBM method for our prediction task. While RNNs and LSTMs are widely used, they present challenges. RNNs are susceptible to the vanishing gradient problem, which occurs due to the activation function, and the exploding gradient problem, which arises from matrix multiplication. The data we work with are highly variable and depend on individual behaviours, making them inherently random. Although LSTMs attempt to handle this randomness by incorporating weighted past data through their gates, designing and stabilizing the weights for such unpredictable data can be difficult.

Given these challenges, we decided to first assess the LightGBM (version 4.4) framework for our prediction task. In the LightGBM model, the data were split as follows: 70% for training (42 datasets out of 60) and 30% for testing (18 out of 60).

## 4. Results and Discussion

### 4.1. Prediction Using Actual Data

First, we present the results of predictions made using only the actual datasets obtained from the in situ experiment. There were five datasets for each sex, but since this number is too small to construct a reliable prediction system with the LightGBM framework, we combined the datasets without considering gender in order to evaluate the accuracy of the prediction system. In this case, the hyperparameters were set as follows: 50% of the data (5 out of 10 datasets) was used for training, while the remaining 50% (5 out of 10) was used for testing. The decision tree depth (*max_depth*) was set to 9, and the *random*_*seed* was fixed. Figure 18 presents a comparison between the predicted and actual values. The results show an *R*^2^ value of 0.656 and a root mean squared error (RMSE) of 5.137, indicating relatively poor prediction accuracy due to the limited dataset. To improve accuracy, it was clear that augmented datasets, as described in Section 3.3, would be necessary.

### 4.2. Setting Hyperparameters of LightGBM

For accurate predictions of water droplet volume using augmented datasets, it is critical to carefully set the hyperparameters of the model. In this study, we focused on adjusting the depth of the decision tree, controlled by the ‘*max_depth*’ parameter. If the number of nodes is too small, multiple predicted data points will converge into one node, resulting in inaccurate predictions. Therefore, an adequate number of nodes is needed to properly distribute the data. In our case, we had a total of 60 datasets, with predictions made on 18 testing datasets. To classify each testing dataset into one node, at least five nodes were required. We examined three different *max_depth* values—3, 5, and 9—and compared the predicted values to the actual data. Figure 19, Figure 20 and Figure 21 show the results for *max_depth* values of 3, 5, and 9, respectively.

For *max_depth* = 3 (Figure 19), the model grouped multiple actual data points into a single predicted data point, resulting in four incorrect predictions. The decision tree diagram for this case, with only three nodes, is shown in Figure 22. When *max_depth* was increased to 5 (Figure 20), seven incorrect predictions occurred, indicating insufficient classification. With *max_depth* = 9 (Figure 21), only three incorrect predictions were made, with a more suitable distribution of nodes. Decision tree diagrams for *max_depth* values of 5 and 9 are shown in Figure 23 and Figure 24, respectively. Based on these results, we determined that a *max_depth* of 9 provided the most accurate predictions and should be adopted depending on the number of test cases.

### 4.3. Differences in Prediction Accuracy with Augmented Data

The use of only five real-world data points was insufficient for accurate predictions. Additionally, continuous data collection beyond these five days was not feasible due to privacy concerns in the restroom facilities. To address this limitation, we augmented the data using two methods, as described earlier. Figure 25, Figure 26 and Figure 27 display the prediction results using augmented datasets generated by the averaging and scaling method (including temperature and humidity), the averaging and scaling method (without temperature and humidity), and the random number generation method, respectively. In these cases, the *random_seed* was fixed to ensure consistency in the training and testing data across all methods. Similarly, *max_depth* was fixed at 9 for all comparisons.

Using the augmented datasets generated by the averaging and scaling method (including temperature and humidity), the model achieved an *R*^2^ value of 0.913 and an RMSE of 1.459 (Figure 25). When the augmented datasets did not include temperature and humidity data, the *R*^2^ value decreased to 0.812, with an RMSE of 2.138 (Figure 26). Finally, using the random number generation method resulted in an *R*^2^ of 0.462 and an RMSE of 4.358, as shown in Figure 27. From Figure 25 and Figure 26, we observed that omitting temperature and humidity slightly decreased prediction accuracy, suggesting that these weather variables are important for more precise predictions. However, for a simplified prediction system, omitting these variables may be acceptable, as their influence on water droplet volume is relatively minor, as shown in Figure 28. This figure compares actual water droplet data with temperature and humidity measurements taken from a male restroom. Figure 29 further illustrates the contribution of various feature variables to the prediction model, showing that temperature and humidity are less significant contributors.

The prediction using the random number generation method resulted in unnatural data, as seen in Figure 27, and produced the least accurate predictions. Therefore, the random number generation method is not suitable for this application.

In this study, we used the DecisionTreeRegressor framework within the LightGBM method for predicting water droplet volumes. LightGBM, which utilises decision trees, is well-suited for time-series data and is commonly used for regression tasks, such as the one in this study.

The main focus of our prediction system was to determine when water droplet volumes would exceed a predefined threshold, indicating maximum dirtiness. This system quantifies dirtiness using water droplet volumes around the washbowl and is designed for practical use in cleaning environments. However, due to the limited amount of real-world data, data augmentation was essential to improve prediction accuracy. Ultimately, our augmented dataset-based model allowed for reliable predictions of water droplet volumes at specific times based on earlier measurements in restrooms.

## 5. Conclusions

In this study, we predicted restroom cleanliness by evaluating the volume of water droplets outside washbowls as an indicator of dirtiness. Using a near-infrared camera, we captured photographs of the water droplets, converted the image data into volume measurements, and accumulated the data over a 1 h period. These measurements were collected over five days through experiments, but the limited amount of data was insufficient to develop an effective learning and prediction algorithm. To address this, we augmented the dataset using two methods: the first method involved averaging and scaling the data, while the second used random number generation. The random number method was found unsuitable, as it caused the water droplet data to fluctuate unpredictably. In contrast, the averaging and scaling method proved effective for our application.

The LightGBM-based prediction system, with optimised hyperparameters and data augmentation, is nearly practical for real-world use. Specifically, this methodology is suitable for restrooms with limited user traffic, such as those in offices and schools. By predicting future dirtiness levels, the system can determine optimal cleaning times or adjust cleaning frequency, ultimately leading to more efficient and future-oriented cleaning practices.

## 6. Future Study

Our system successfully predicts water droplet volume 1 h ahead, but it could also predict 2 or 3 h future volumes. The process is sequential: we first predicted the 1 h ahead droplet volume, used this as actual data, and then predicted the 2 h ahead volume, followed by 3 h prediction. However, the accuracy of predictions for the 2 and 3 h intervals is expected to be lower than for the 1 h prediction. To improve the accuracy of longer-term predictions, additional data collection is necessary.

Currently, our prediction system relies on photographs of water droplets, which may not be practical in all real-world scenarios. Future work will explore methods that eliminate the need for photographs, making the system more feasible for actual cleaning operations.

Although the LightGBM method treats data as an array, not a time-series, other techniques could be integrated into our system. LSTM networks, which are designed to handle time-series data and retain past information, could enhance the prediction accuracy. Furthermore, the GRU [36,37] offers a simpler and computationally less intensive alternative to LSTM. RNNs are another promising approach. We plan to evaluate these three methods—LSTM, GRU, and RNN—to determine the most optimal solution for our application.

## Figures and Tables

**Figure 1 sensors-25-02186-f001:**
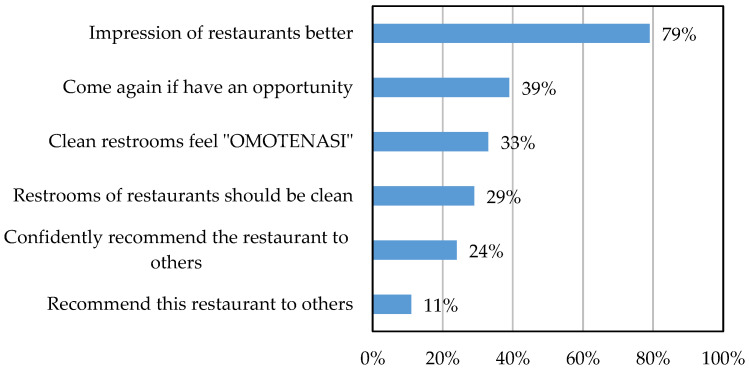
Responses from 2000 participants to the survey question: ‘How do you feel about clean restrooms in restaurants?’ [6].

**Figure 2 sensors-25-02186-f002:**
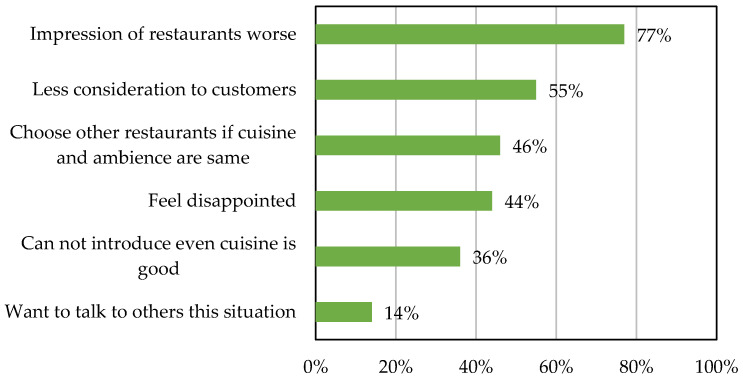
Responses from 2000 participants to the survey question ‘How do you feel about dirty restrooms in restaurants?’ [6].

**Figure 3 sensors-25-02186-f003:**
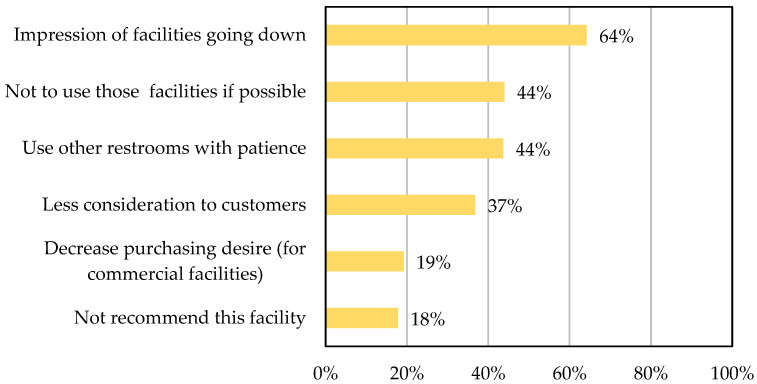
Responses of participants to the survey question ‘How comfortable do you feel using public restrooms?’ [7].

**Figure 4 sensors-25-02186-f004:**
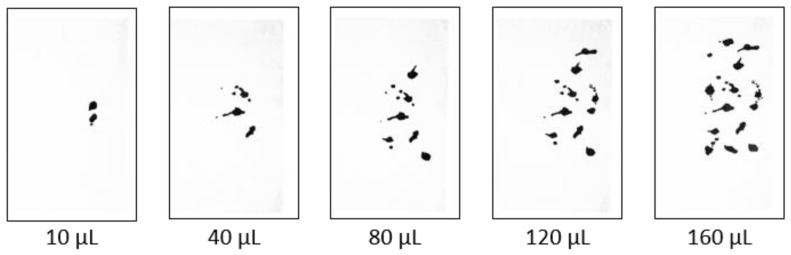
Relationship between pixel images of water droplets dropped using a digital pipette and the corresponding volumes.

**Figure 5 sensors-25-02186-f005:**
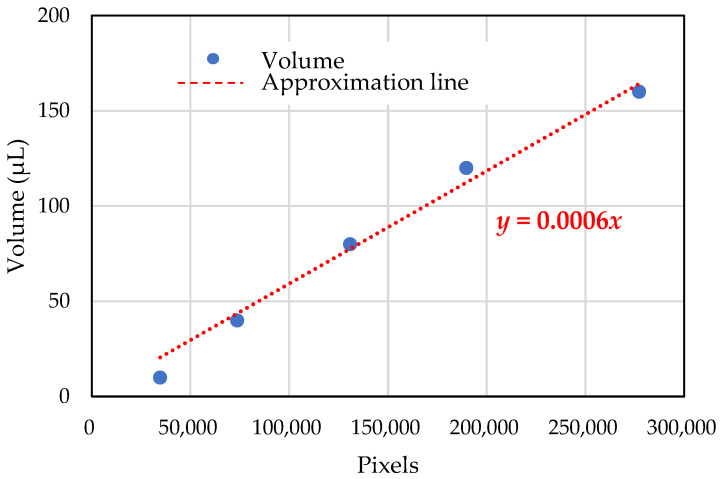
Conversion of pixel data to volume from the collected images.

**Figure 6 sensors-25-02186-f006:**
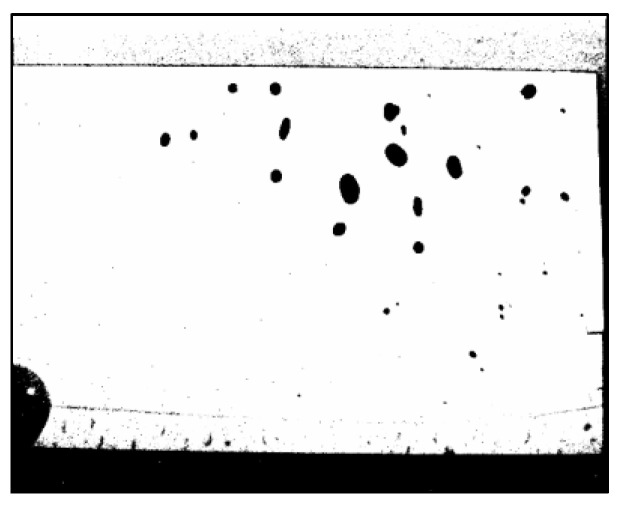
Example of a bitmap image of a water droplet captured by a near-infrared camera.

**Figure 7 sensors-25-02186-f007:**
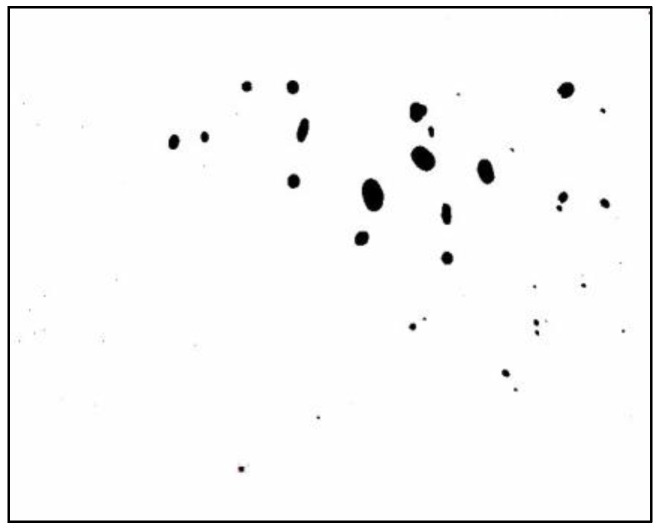
Bitmap image after removing the irrelevant black areas from Figure 6.

**Figure 8 sensors-25-02186-f008:**
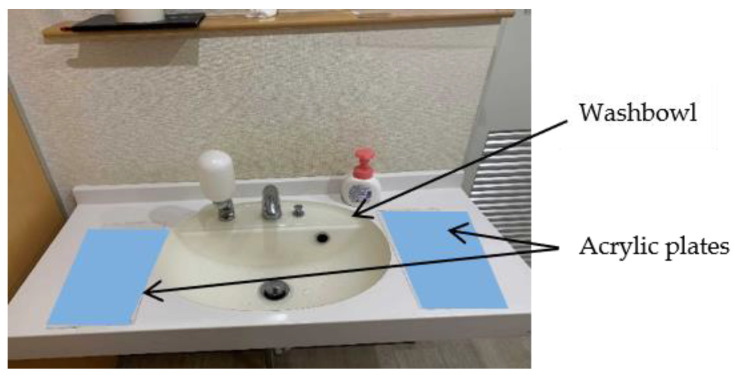
Photograph of the washbowl with acrylic plates placed on either side. Note that the acrylic plates, though transparent in the experiment, are shown in blue here for clarity.

**Figure 9 sensors-25-02186-f009:**
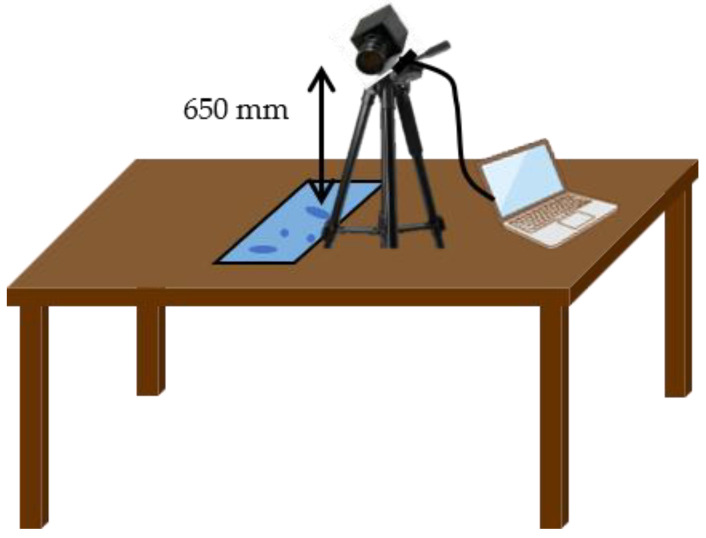
Setup of the camera was set up in a different room from the restroom to capture photographs of water droplets on the acrylic plates.

**Figure 10 sensors-25-02186-f010:**
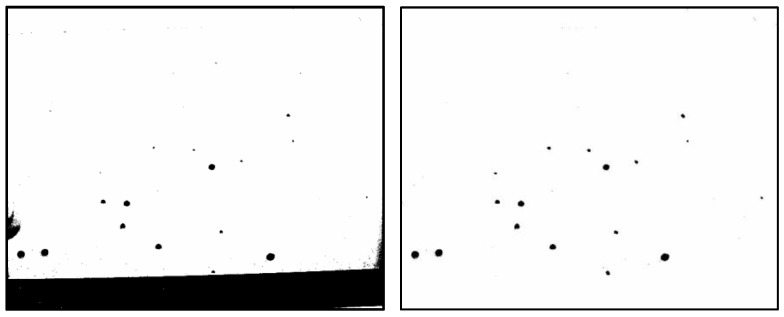
Captured bitmap image of water droplets taken at 10:30 on the first day of the experiment at male restroom (**left**) and the processed image after removing irrelevant black areas (**right**).

**Figure 11 sensors-25-02186-f011:**
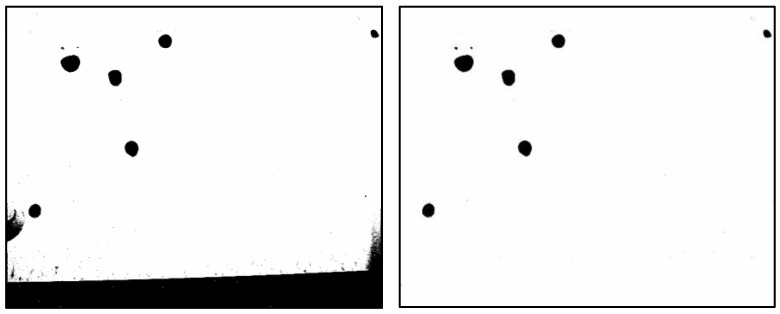
Captured bitmap image of water droplets taken at 11:30 on the first day of the experiment at male restroom (**left**) and the processed image after removing irrelevant black areas (**right**).

**Figure 12 sensors-25-02186-f012:**
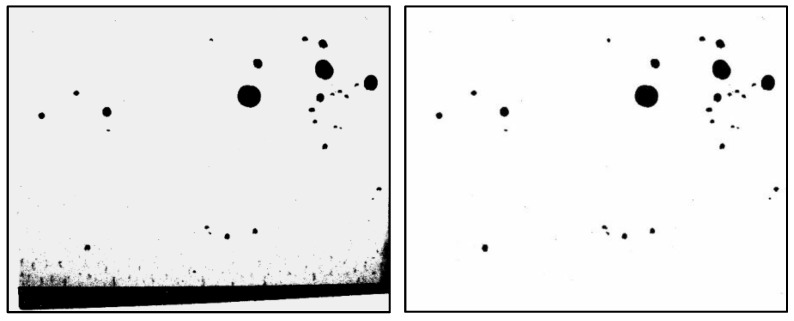
Captured bitmap image of water droplets taken at 12:30 on the first day of the experiment at male restroom (**left**) and the processed image after removing irrelevant black areas (**right**).

**Figure 13 sensors-25-02186-f013:**
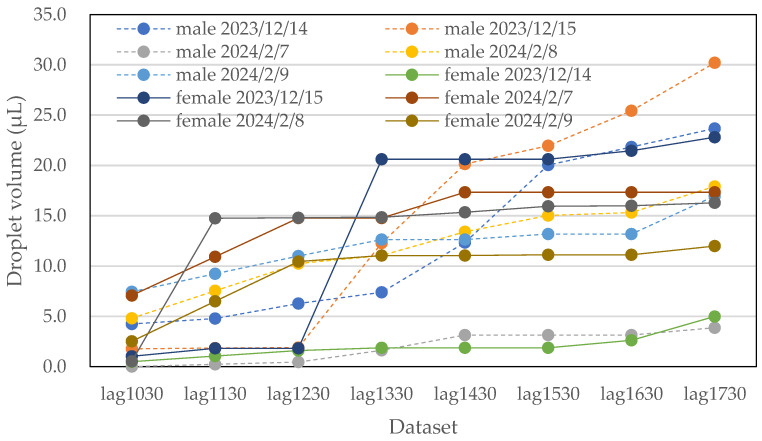
Five-day data from male (dashed lines) and female (solid lines) restrooms, converted to water droplet volume data.

**Figure 14 sensors-25-02186-f014:**
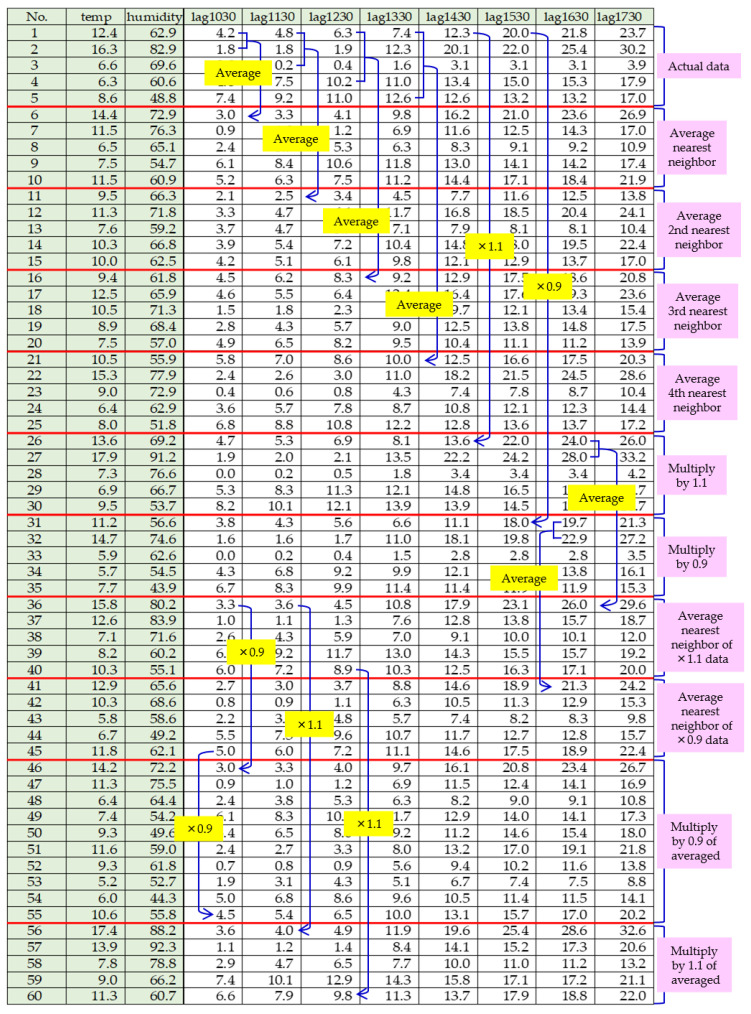
Data augmentation results using the average and multiply method.

**Figure 15 sensors-25-02186-f015:**
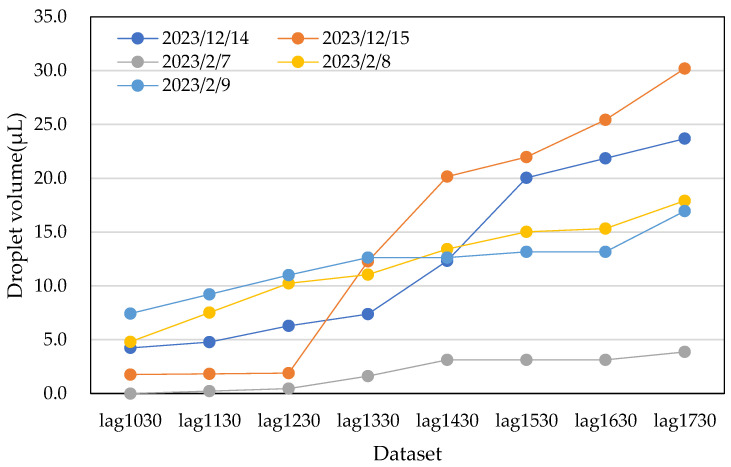
Actual water droplet volume data from the male restroom over five days, excerpted from Figure 13.

**Figure 16 sensors-25-02186-f016:**
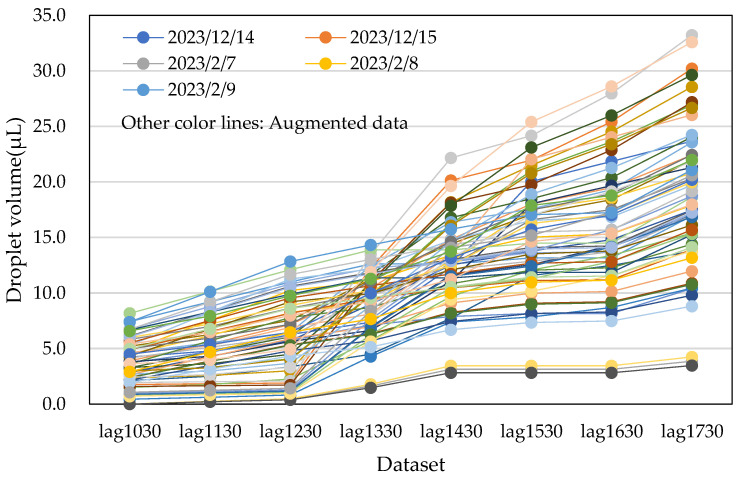
Augmented water droplet volume data from the male restroom using the averaging and scaling method. The total number of datasets is 60, including the five actual datasets.

**Figure 17 sensors-25-02186-f017:**
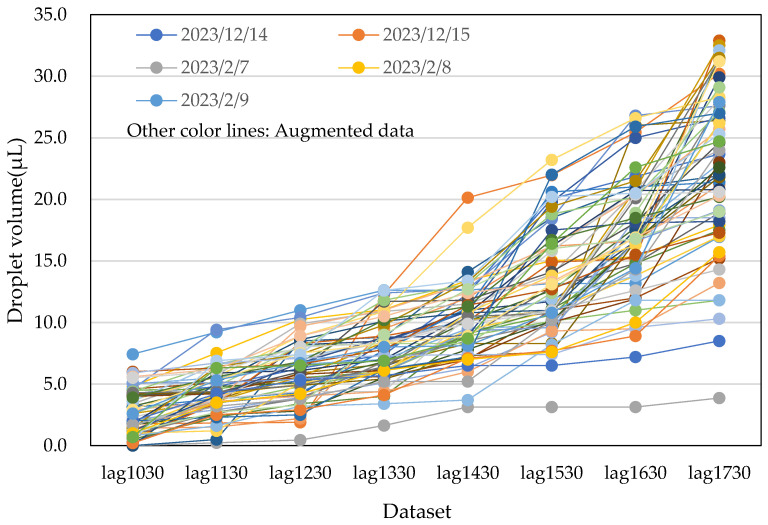
Augmented water droplet volume data from the male restroom using the random number method. The total number of datasets is 60, including five actual datasets.

**Figure 18 sensors-25-02186-f018:**
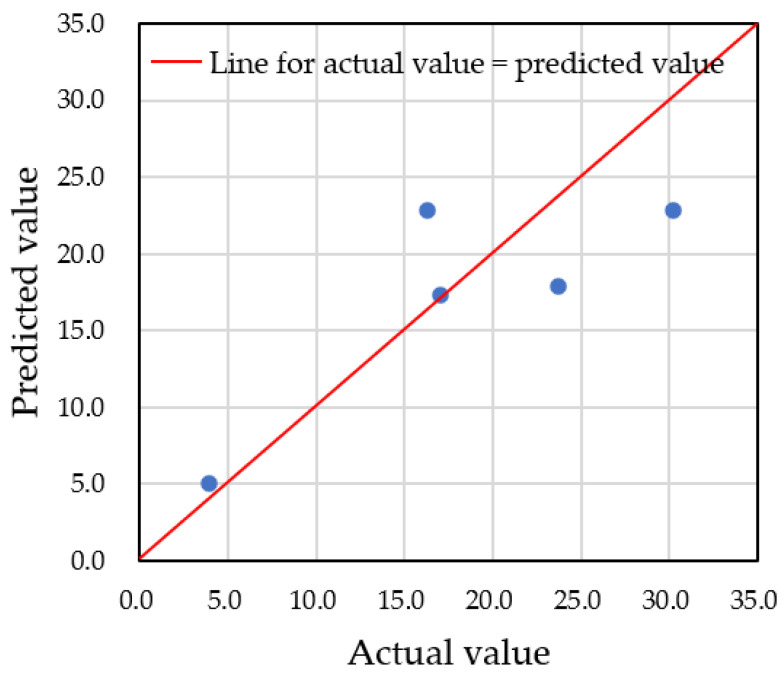
Comparison of predicted values and actual values when only the actual datasets were used for prediction.

**Figure 19 sensors-25-02186-f019:**
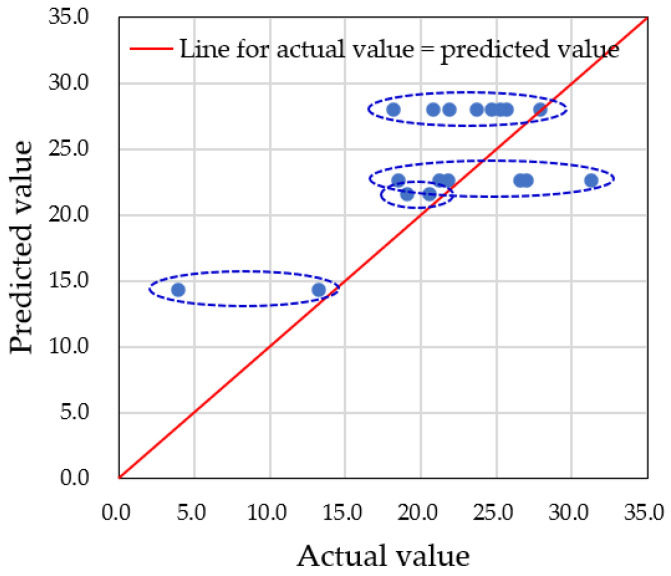
Four multiple predictions were obtained when the *max_depth* was set to 3. Points having the same predicted value for different actual values are grouped with dotted line circle.

**Figure 20 sensors-25-02186-f020:**
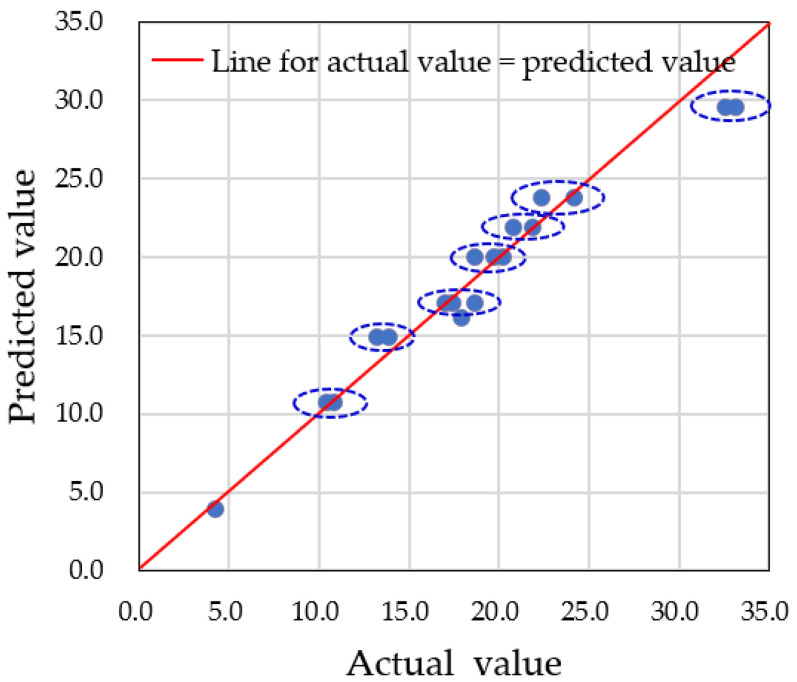
Seven multiple predictions were obtained when the *max_depth* was set to 5. Points having the same predicted value for different actual values are grouped with dotted line circle.

**Figure 21 sensors-25-02186-f021:**
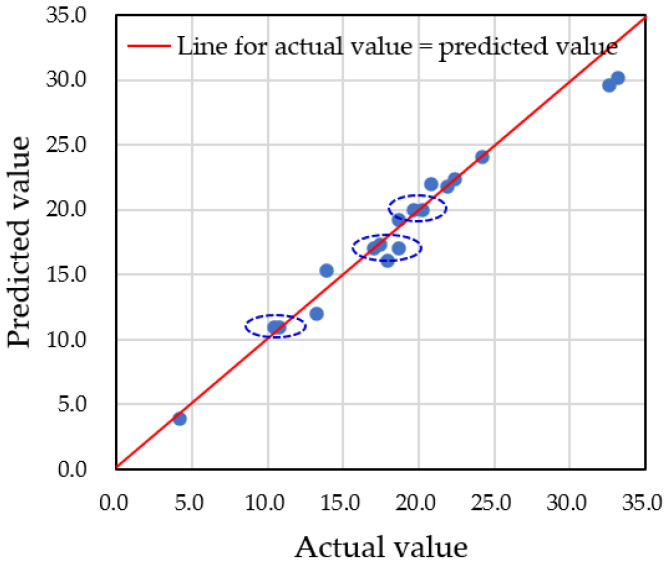
Three multiple predictions were obtained when the *max_depth* was set to 9. Points having the same predicted value for different actual values are grouped with dotted line circle.

**Figure 22 sensors-25-02186-f022:**
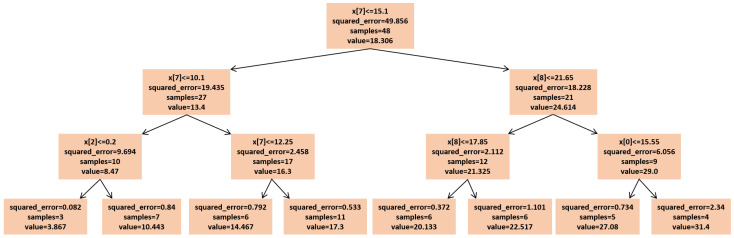
Decision tree diagram for *max_depth* = 3.

**Figure 23 sensors-25-02186-f023:**
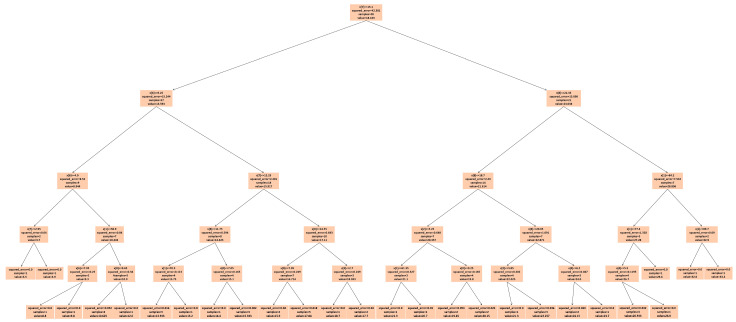
Decision tree diagram for *max_depth* = 5.

**Figure 24 sensors-25-02186-f024:**
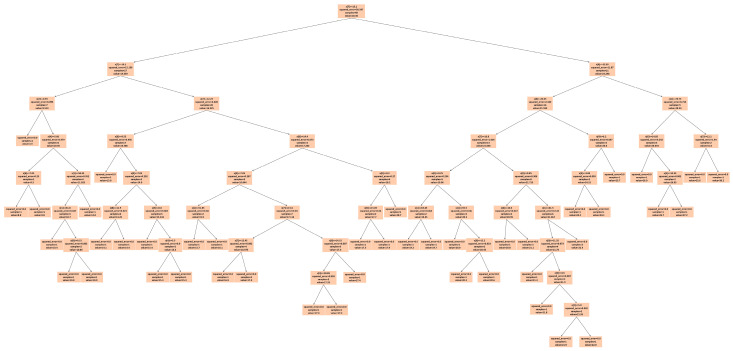
Decision tree diagram for *max_depth* = 9.

**Figure 25 sensors-25-02186-f025:**
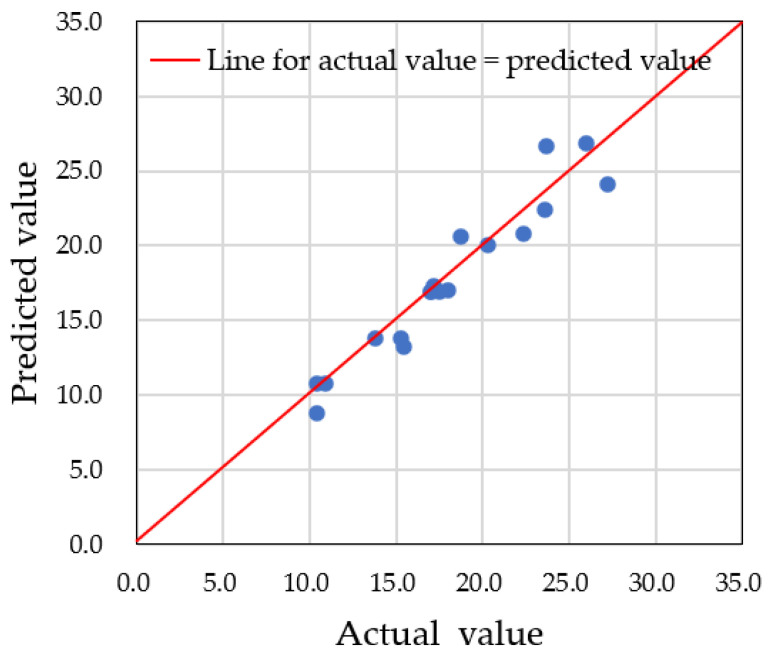
Comparison of predicted values and actual values using augmented datasets generated by the averaging and scaling method, including temperature and humidity.

**Figure 26 sensors-25-02186-f026:**
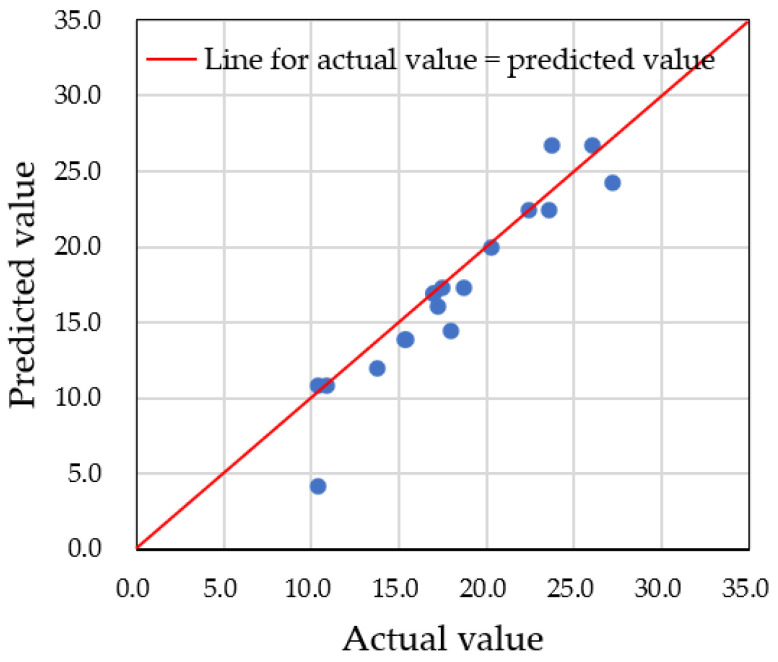
Comparison of predicted values and actual values using augmented datasets generated by the averaging and scaling method, excluding temperature and humidity.

**Figure 27 sensors-25-02186-f027:**
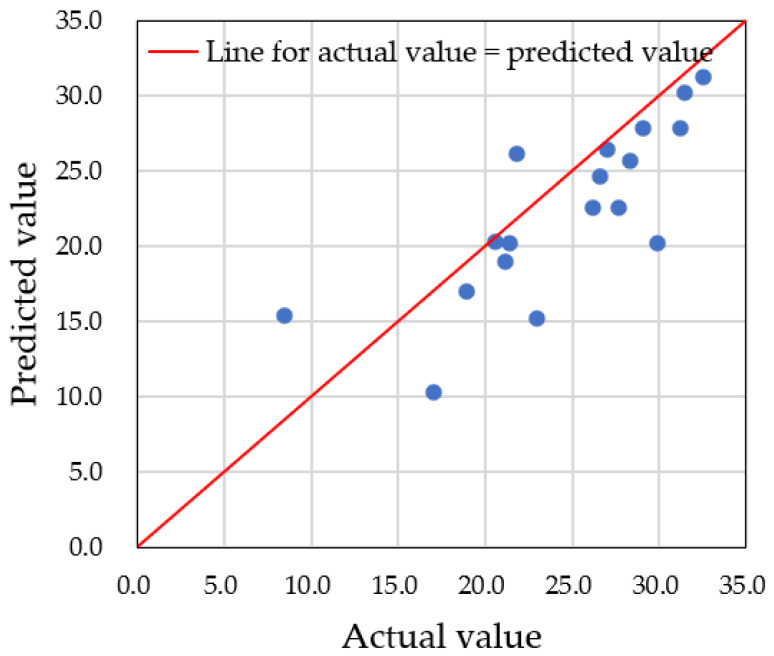
Comparison of predicted values and actual values using augmented datasets generated by the random number generation method.

**Figure 28 sensors-25-02186-f028:**
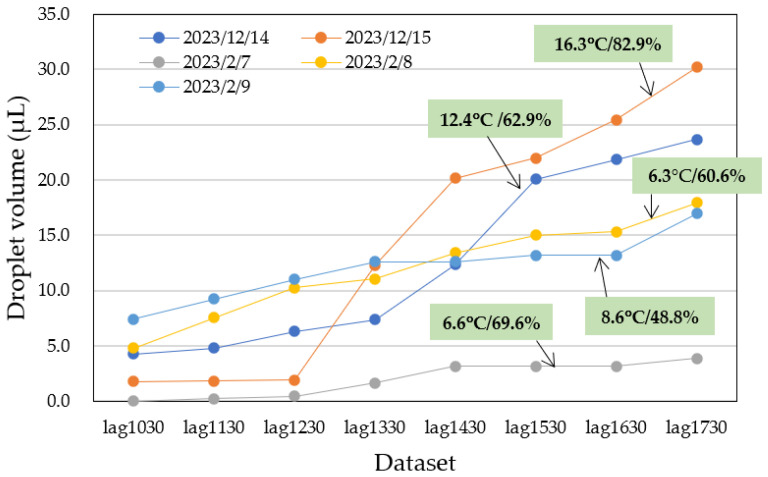
Actual water droplet volume data collected from the male restroom, along with temperature and humidity measurements.

**Figure 29 sensors-25-02186-f029:**
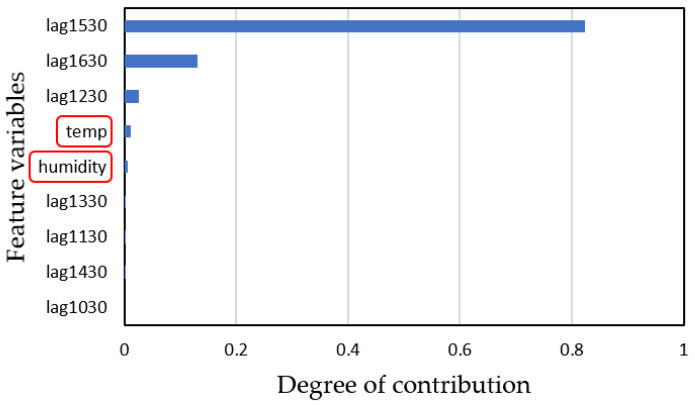
Contribution of feature variables in predicting water droplet volume. Temperature and humidity red boxed in are less significant contributors among feature variables.

**Table 1 sensors-25-02186-t001:** Specifications of the camera used in the experiments [10].

Image Sensor	InGaAs
Number of pixels	640 × 512 pixels
Wavelength sensitivity	0.9–1.7 µm
Frame rate	98 fps
Camera dimensions	61 × 59 × 81 mm (width × height × depth) (without the lens)

**Table 2 sensors-25-02186-t002:** Five-day data collected from both male and female restrooms, converted to water droplet volumes, along with temperature and humidity data.

Sex	Temperature (°C)	Humidity (%)	lag1030 (μL)	lag1130 (μL)	lag1230 (μL)	lag1330 (μL)	lag1430 (μL)	lag1530 (μL)	lag1630 (μL)	lag1730 (μL)
Male	12.4	62.9	4.2	4.8	6.3	7.4	12.3	20.0	21.8	23.7
Male	16.3	82.9	1.8	1.8	1.9	12.3	20.1	22.0	25.4	30.2
Male	6.6	69.6	0.0	0.2	0.4	1.6	3.1	3.1	3.1	3.9
Male	6.3	60.6	4.8	7.5	10.2	11.0	13.4	15.0	15.3	17.9
Male	8.6	48.8	7.4	9.2	11.0	12.6	12.6	13.2	13.2	17.0
Female	12.4	62.9	0.5	1.0	1.6	1.9	1.9	1.9	2.6	5.0
Female	16.3	82.9	1.0	1.8	1.8	20.6	20.6	20.6	21.5	22.8
Female	6.6	69.6	7.1	10.9	14.8	14.8	17.3	17.3	17.3	17.3
Female	6.3	60.6	0.5	14.7	14.8	14.8	15.3	15.9	16.0	16.3
Female	8.6	48.8	2.5	6.5	10.5	11.0	11.0	11.1	11.1	12.0

**Table 3 sensors-25-02186-t003:** Data augmentation results using the random number generation method. Dataset from No. 1 to 5 are actual data and others are augmented datasets.

No.	lag1030	lag1130	lag1230	lag1330	lag1430	lag1530	lag1630	lag1730
1	4.2	4.8	6.3	7.4	12.3	20.0	21.8	23.7
2	1.8	1.8	1.9	12.3	20.1	22.0	25.4	30.2
3	0.0	0.2	0.4	1.6	3.1	3.1	3.1	3.9
4	4.8	7.5	10.2	11.0	13.4	15.0	15.3	17.9
5	7.4	9.2	11.0	12.6	12.6	13.2	13.2	17.0
6	4.5	4.6	5.5	6.0	9.1	10.6	16.5	26.2
7	3.2	3.4	8.2	8.5	11.0	11.7	16.6	29.9
8	4.6	5.1	5.5	5.6	7.0	11.0	12.0	15.2
9	0.3	2.6	3.9	6.4	8.9	12.8	14.8	18.9
10	1.2	3.7	5.9	6.6	8.3	8.3	26.0	26.4
11	0.0	0.5	8.5	8.8	14.1	18.5	21.0	21.9
12	3.2	4.8	5.1	6.8	10.1	12.6	17.7	31.3
13	4.6	9.4	10.4	12.4	12.7	18.4	26.8	27.6
14	1.4	1.5	2.2	9.5	12.0	16.2	16.6	21.2
15	3.4	5.7	9.9	10.9	11.5	13.9	20.6	24.0
16	1.5	4.1	5.0	6.1	9.4	10.9	13.8	17.0
17	5.0	5.5	6.6	7.0	9.5	9.8	17.6	22.9
18	1.9	3.4	4.8	6.0	8.9	10.1	11.0	11.8
19	0.7	3.5	4.1	8.4	10.9	19.8	25.0	26.6
20	0.6	3.4	8.3	8.9	10.1	14.9	15.2	32.9
21	0.9	2.7	3.8	6.8	8.5	10.0	16.6	27.7
22	2.6	3.6	5.6	8.4	10.2	11.0	20.8	32.5
23	3.1	3.8	4.3	7.7	11.2	20.6	21.0	21.4
24	0.5	2.3	3.4	4.0	9.1	11.1	14.8	17.5
25	2.1	2.9	3.9	5.1	6.3	9.2	18.5	18.5
26	2.9	3.6	9.7	11.0	13.2	16.2	20.3	25.7
27	2.3	3.8	4.4	4.4	10.5	15.8	20.7	28.1
28	3.7	5.6	8.9	10.9	17.7	23.2	26.6	28.3
29	3.6	6.6	7.2	8.4	10.3	11.9	13.8	32.1
30	2.6	4.8	8.1	8.3	8.3	12.4	18.9	29.1
31	1.8	5.9	6.1	7.0	10.7	16.2	20.7	20.8
32	4.0	4.2	5.8	6.1	7.1	10.0	11.9	23.0
33	4.1	4.3	6.0	11.7	11.8	14.1	18.1	24.6
34	4.2	4.3	4.9	5.4	7.3	7.6	16.6	21.8
35	0.0	5.1	8.6	10.1	10.8	11.0	17.5	22.0
36	1.7	2.5	2.8	5.5	7.8	10.1	16.6	22.6
37	5.1	5.1	5.4	5.9	9.3	9.6	16.6	19.1
38	1.3	3.6	4.0	4.4	6.0	9.3	9.5	13.2
39	1.6	3.7	7.0	8.4	10.1	11.0	12.6	14.3
40	1.0	1.2	5.1	5.9	10.8	13.8	16.3	26.2
41	0.8	1.6	3.2	3.4	3.7	8.2	11.8	11.8
42	4.4	6.6	7.9	11.8	13.1	18.8	20.2	27.1
43	1.2	4.0	6.7	8.8	8.9	17.5	18.1	18.2
44	6.0	6.3	6.6	7.8	11.4	12.7	15.5	17.3
45	4.3	4.4	4.9	6.4	10.5	10.7	20.1	31.3
46	1.4	6.4	7.4	9.8	13.3	19.4	21.5	31.5
47	2.0	2.3	2.5	6.2	7.2	22.0	25.9	27.0
48	3.9	5.6	6.8	10.2	11.3	16.7	18.5	20.2
49	5.9	6.7	7.1	7.3	7.4	7.4	9.6	10.3
50	5.6	6.2	8.9	10.5	12.4	13.3	16.9	20.3
51	5.5	6.0	8.0	8.3	9.9	10.9	20.4	20.6
52	2.9	5.3	7.5	8.3	8.9	13.1	16.9	31.2
53	0.4	6.9	7.4	12.6	13.4	20.2	20.5	25.3
54	1.1	2.7	3.3	9.0	12.7	16.0	16.8	19.0
55	2.0	4.2	5.3	5.7	6.5	6.5	7.2	8.5
56	0.2	2.5	2.9	4.1	7.1	7.7	8.9	15.4
57	1.7	3.2	3.8	5.2	5.2	10.2	14.3	24.0
58	1.0	3.5	4.2	6.1	7.0	7.6	10.0	15.7
59	2.6	5.3	6.7	8.0	8.1	10.8	14.4	27.9
60	0.7	6.3	6.5	6.9	8.7	16.4	22.6	24.7

## Data Availability

Data are contained within the article.

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
