# Peer review of "Predicting Restroom Dirtiness Based on Water Droplet Volume Using the LightGBM Algorithm"

_sensors, 2025, doi:10.3390/s25072186_

Round 1

Reviewer 1 Report

Comments and Suggestions for Authors

The paper uses an algorithm to measure the volume of water droplets to estimate the cleanliness of the toilet is very interesting, and there are the following problems

(1) The proposed method is not described in detail in the paper, and it is recommended that the method and the experiment be described separately

(2) How to measure water droplets in the method is not specifically given, and where are the innovation points

(3) There is a lack of visual representation of the experimental results

(4) The line chart from Figure 13 to Figure 15 is not clear to see, and the points and lines are too large

Comments on the Quality of English Language

The English could be improved to more clearly express the research.

Reviewer 2 Report

Comments and Suggestions for Authors

The article proposes a method of evaluating restroom dirtiness. However, I did not find interesting findings from their case studies. Moreover, why the water droplets connect to the restroom's dirtiness should be explained more clearly. If the whole quality of the paper could be improved, I would like to review and accept it.

In the Introduction Section, the total number of respondents in those tables is suggested to be listed.

The quality of figures and tables should be improved, for instance:

1. How did you get the Figure 4? Do you have any references or experiments?

2. Some data were hidden in Figures 12 and 20. Please show them.

3. Figures 14 and 15 are in a mass. Moreover, no legend explains the meaning of each color from Fig 13 to Fig 15.

Comments on the Quality of English Language

The quality of English could be improved.

Round 2

Reviewer 2 Report

Comments and Suggestions for Authors

The authors have answered all of the reviewer's comments. The English of the whole paper has been improved. The resolutions of Figures 22-24 must be increased for the publication. They are hard to read in the correct version.